# Emerging Role of Cancer-Associated Fibroblasts in Progression and Treatment of Hepatocellular Carcinoma

**DOI:** 10.3390/ijms24043941

**Published:** 2023-02-15

**Authors:** Hikmet Akkız

**Affiliations:** Department of Gastroenterology and Hepatology, The University of Bahçeşehir, Istanbul 34425, Turkey; hakkiz@superonline.com; Tel.: +90-322-458-23-01

**Keywords:** cancer-associated fibroblasts, tumor microenvironment, tumor-associated neutrophils, crosstalk, hepatocellular carcinoma

## Abstract

Hepatocellular carcinoma (HCC) is one of the most prevalent cancers worldwide and the fourth leading cause of cancer-related death globally. Tumor cells recruit and remodel various types of stromal and inflammatory cells to form a tumor microenvironment (TME), which encompasses cellular and molecular entities, including cancer-associated fibroblasts (CAFs), tumor-associated macrophages (TAMs), tumor-associated neutrophils (TANs), immune cells, myeloid-derived suppressor cells (MDSCs), immune checkpoint molecules and cytokines that promote cancer cell growth, as well as their drug resistance. HCC usually arises in the context of cirrhosis, which is always associated with an enrichment of activated fibroblasts that are owed to chronic inflammation. CAFs are a major component of the TME, providing physical support in it and secreting various proteins, such as extracellular matrices (ECMs), hepatocyte growth factor (HGF), insulin-like growth factor 1/2 (ILGF1/2) and cytokines that can modulate tumor growth and survival. As such, CAF-derived signaling may increase the pool of resistant cells, thus reducing the duration of clinical responses and increasing the degree of heterogeneity within tumors. Although CAFs are often implicated to be associated with tumor growth, metastasis and drug resistance, several studies have reported that CAFs have significant phenotypic and functional heterogeneity, and some CAFs display antitumor and drug-sensitizing properties. Multiple studies have highlighted the relevance of crosstalk between HCC cells, CAFs and other stromal cells in influence of HCC progression. Although basic and clinical studies partially revealed the emerging roles of CAFs in immunotherapy resistance and immune evasion, a better understanding of the unique functions of CAFs in HCC progression will contribute to development of more effective molecular-targeted drugs. In this review article, molecular mechanisms involved in crosstalk between CAFs, HCC cells and other stromal cells, as well as the effects of CAFs on HCC-cell growth, metastasis, drug resistance and clinical outcomes, are comprehensively discussed.

## 1. Introduction

Hepatocellular carcinoma (HCC), which accounts for roughly 90% of primary liver cancer, is one of the most prevalent and lethal cancers worldwide, ranking as sixth in incidence and fourth in mortality [1,2]. Despite implementation of a vaccine against the hepatitis B virus (HBV) worldwide and the introduction of antiviral drugs that provide a cure for hepatitis C virus (HCV) infection, the incidence of HCC has been rising dramatically, particularly in Western countries, over the last decade [1,2,3]. HCC usually arises in the setting of liver cirrhosis caused by HBV infection, HCV infection, heavy alcohol consumption or nonalcoholic fatty liver disease (NAFLD) [2,3,4,5,6]. According to WHO data, HBV infection affects more than 250 million individuals worldwide, and almost 1 million die annually of complications of cirrhosis and HCC [1,2,3]. HBV infection is the most frequent risk factor for development of HCC, accounting for about 50% of cases [1,2,3]. The incidence of HCV-related HCC has substantially decreased, owing to patients who achieved virological cures [2,3,4]. NAFLD is the most prevalent chronic liver disease, affecting roughly 25% of the global population, and is emerging as a leading cause of liver cirrhosis and HCC [5]. NAFLD contains a spectrum of liver pathologies that range from steatosis to nonalcoholic steatohepatitis (NASH), which is the fastest-growing cause of HCC in Europe and the USA [7,8].

Precision-medicine strategies created through groundbreaking technological advances of the 21st century have been transformative in cancer treatment, moving away from drugs that target tumors broadly, such as chemotherapy and radiotherapy, and toward implementation of targeted drugs that modulate the immune response against tumors [9]. These strategies are particularly effective when they directly target genetically activated oncogenes that cause aberrant kinase signaling, such as BCR-ABL and EML4-ALK fusions that arise from different molecular mechanisms [9,10]. However, such targeted treatments are often followed by drug resistance, resulting in cancer recurrence [9,10,11]. Despite immense advances in imaging technologies, molecular-targeted drugs and surgical techniques in the past decade, the overall survival rate of patients with HCC remains dismal, and only in early-stage HCC can surgical techniques, such as hepatic resection and liver transplantation or local/regional treatments, provide cures [10,11,12,13]. However, the recurrence rate is extremely high in advanced HCC following surgery or local/regional treatments [12,13]. In clinical practice, the vast majority of HCCs are diagnosed at an advanced stage where systemic therapies, such as multityrosine kinase inhibitors and immunotherapies, are the only treatment options [8,12]. Precision medicine has provided a paradigm shift in the treatment of advanced-stage HCC patients in the last decade [9,11]. Immune checkpoint inhibition with atezolizumab and bevacizumab, an antivascular endothelial growth factor neutralizing antibody, has become a first-line therapy for patients with advanced HCC [12,14]. All of these developments indicate that it is crucial to better understand the underlying molecular mechanisms that drive HCC progression and metastasis in order to develop more effective therapies and determine the dynamics that influence treatment consequences.

A tumor microenvironment (TME) comprises complex structures and functions and has significant impacts on tumor growth, energy metabolism and progression [12,15,16]. The TME contains cellular and molecular components, including cancer-associated fibroblasts (CAFs), tumor-associated macrophages (TAMs), tumor-associated neutrophils (TANs), regulatory T cells (Tregs), myeloid-derived suppressor cells (MDSCs), immune cells, inhibitory cytokines, immune checkpoint receptors and ligands and an extracellular matrix (ECM) [9,10,12,15,16,17,18]. CAFs are a key component of the stroma and critically modulate cancer progression through various mechanisms, including production of growth factors, inflammatory ligands and exosomes as well as ECM remodeling, angiogenesis influencing, tumor mechanics and treatment responses [15,16,17]. More than 80% of HCCs arise in the cirrhotic liver, accompanied with activation, proliferation and accumulation of fibroblasts [2,3,4,5,8]. As such, CAFs enable an efficient microenvironment for HCC occurrence, progression and metastasis [11,15,16]. During hepatocarcinogenesis, HCC cells produce various molecules that recruit CAFs within the TME [10,12,15]. CAFs also secrete multiple soluble factors, such as growth factors, inflammatory ligands, angiogenic factors and chemokines, that induce tumor-cell proliferation and metastasis [11,12,15,16,17,18,19]. Their crosstalk with cancer cells is mediated with a complex signaling network that consists of signaling pathways for transforming growth factor-beta (TGF-β), mitogen-activated protein kinase (MAPK), Wnt/β-catenin, Janus kinase/signal transducers and activators of transcription (JAK/STATs), epidermal growth factor receptor (EGFR), nuclear factor-light kappa-enhancer of activated B cells (NF-κB), etc. [20,21,22]. Recently, experimental studies that investigated the role of CAFs in HCC progression revealed that CAFs secrete various chemokines within the TME, promoting HCC-cell invasion and metastasis through activation of either the Hedgehog or TGF-β pathway [16,23]. The TGFβ pathway plays a dual role, being both a tumor suppressor in premalignant cells and a tumor promoter in cancer cells [21,22]. The TGFβ/Smad pathway mediates EMTs through Snail/Slug expression in tumor endothelial cells to support sprouting angiogenesis and accumulation of myofibroblasts and CAFs in a TME [21,22]. Emerging evidence has demonstrated that the CAF-mediated JAK/STAT signaling pathway is widely involved in cancers through various tumor biological processes, including increased cell plasticity, proliferation, migration, EMT, angiogenesis and metastasis [21,22,23]. In HCC, CAF-derived IL-6 promotes HCC-cell EMT, which in turn activates the IL-6/JAK/STAT3 pathway to induce expression of TG2 for acquisition of EMT phenotypes [20,22,23]. Although growing evidence indicates the significant role of CAFs in production of secretory molecules, the molecular regulatory mechanisms of CAFs have not yet been completely elucidated due to their diverse biological functions and intricate TME entities [16,17,18,20,21,22].

The ability of CAFs to modulate the immune response has been greatly appreciated over the past few years. HCC-derived CAFs recruit immune cells, such as neutrophils, monocytes and dendritic cells, and promote these cells to acquire immunosuppressive phenotypes that foster immune escape [12,15]. Studies investigating CAFs, through alteration of their numbers, subtypes or functionality, are ongoing to improve cancer therapies. However, trials addressing the crosstalk between CAFs, HCC cells and immune cells face numerous challenges [12,15,16]. The development of novel coculture models that assess CAF biology and implementation of single-cell RNA-sequencing (scRNA-seq) techniques have identified immense levels of CAF heterogeneity in various cancer types [12,16,17,18,19,20]. Although molecular trials enable insights into the nature of CAF heterogeneity, the extent of this heterogeneity and the roles of different CAF subtypes remain largely unknown [17,18,19,20,21]. It is clear that different CAF subsets could exert diverse roles in HCC progression; therefore, targeting CAF populations individually could result in favorable clinical outcomes [17,18,20,21].

## 2. What Is a Fibroblast?

Emerging studies have shown that in a healthy liver, stromal cells constitute connective tissue to provide a supportive framework to the liver [18]. In normal tissue, fibroblasts are considered inactivated mesenchymal cells embedded in the ECM of interstitial fibers [18,24]. They secrete several types of collagens or matrix metalloproteinase (MMP) to maintain the integrity of and remodel the ECM, enabling a balance between the matrix components [10,17,24]. During chronic liver injury, quiescent fibroblasts are activated and transdifferentiated into myofibroblasts (MFBs), which express TGFβ [18,25]. CAFs consist of multiple subtypes involved in the pathogeneses of different diseases [24,25]. The vast majority of fibroblasts are derived from the primitive mesenchyme [10,17,24]. The main difficulty in identification of fibroblasts is the absence of specific markers that are not expressed by other cell types. Fibroblast activation protein (FAP) is produced by most CAFs and normal fibroblasts, indicating a common cell lineage [18,25]. In addition, along with other criteria, such as cell shape and location, vimentin and platelet-derived growth factor receptor-α (PDGFRα) are used to distinguish fibroblasts from other cells. Considering clinical and laboratory data, fibroblasts are characterized through a combination of their morphology; tissue location; and absence of lineage markers for epithelial cells, endothelial cells and leukocytes [17,18,24,25]. In physiological conditions, fibroblasts are the main cells that express ECM, and emerging evidence suggests that this function is altered with age [18]. Fibroblasts also have a critical role in tissue repair and become activated following tissue damage [17]. During inflammatory states and wound healing, they can express TGF-β and transdifferentiate into myofibroblasts (MFBs) that express αSMA. Numerous studies have documented that following liver injury, MFBs engage in interaction with adjacent epithelia and affect local epithelial-stem-cell behavior [24,25]. They can also induce angiogenesis through production of vascular endothelial growth factor A (VEGFA) and regulate immune response through expression of cytokines and chemokines [26]. Fibroblasts also have a structural role within the immune system; fibroblastic reticular cells (FRCs) within lymph nodes form ECM conduits, serving as a migration highway for leukocytes and potential antigens and providing effective immune surveillance [17]. Furthermore, fibroblasts augment immune tolerance through production and presentation of normally tissue-specific antigens [27]. A trial demonstrated an intricate interplay between fibroblastic cells and epithelial cells. For example, hepatic stellate cells (HSCs) are found in livers that store lipid droplets [17,28]. The balance between quiescence and activation of HSCs is orchestrated through vitamin D receptors, the deletion of which results in liver fibrosis [29]. As such, fibroblasts are not simply producers of ECM but exhibit critical roles in connection of many other cell types during both physiological conditions and tissue repair [17].

## 3. Origin and Activation Mechanisms of CAFs

CAFs are a major component of the HCC microenvironment, and activated fibroblasts encompass the main forms of CAF found in some cancers [17,30]. Emerging evidence indicates that CAFs are a heterogeneous population of cells that depend on their numerous cellular precursors [23]. Multiple markers have been found to identify CAFs, including αSMA, fibroblast activation proteins (FAPs), fibroblast specific protein 1 (FSP1), vimentin and PDGF receptors (PDGFRs)-α and β [10,11,17,18]. However, the absence of fibroblast-unique markers generates a challenge in determining the precise cells of origin of CAFs. To better understand the origins of CAFs, many researchers have used mouse models in which cells can be irreversibly labeled using transgenic techniques and well-characterized models of disease progression are available [17,30]. Growing studies have reported that CAFs can originate from pancreatic and hepatic stellate cells (HSCs), neutrophils, bone-marrow-derived mesenchymal stem cells (MSCs), adipocytes, pericytes, endothelial cells and cancer cells that are undergoing an epithelial–mesenchymal transition (EMT) [10,18,30]. However, the majority of these cell types identified to be CAF precursors have been derived from in vitro experiments and bone-marrow transplantation studies [18]. Injection of bone-marrow-derived MSCs into mice has shown that MSCs can transdifferentiate into CAFs [17] Figure 1. 

Neutrophils are the main source of CAFs, and neutrophil-derived CAFs are unique in many aspects [17]. High expression of CAF markers, such as αSMA and FAPα, distinguishes CAFs from neutrophils [18]. Perivascular cell-originated CAFs have been demonstrated to play a significant role in tumor metastasis [10,12]. Light-microscopy examination of CAFs revealed specific morphological properties of these cells; for example, CAFs were identified as cells with larger volumes, richer cytoplasms and serrated nuclei. In electron-microscope examination, abundant rough endoplasmic reticula, free ribosomes, Golgi apparatuses and stress fibers could be detected in CAFs [18]. Neutrophils have a pivotal function in tissue repair and play an important role in protecting cells from necrosis [10]. Unlike neutrophils, CAFs are cells characterized with high proliferation and migration capacity and can remodel the ECM, mediate immune escape and contribute to tumor drug resistance in the TME [31,32]. The TME can promote transdifferentiation of neutrophils into CAFs through various biological factors, such as growth factors, cytokines and chemokines produced by stromal cells [10,28,31,32]. Furthermore, mutations in neutrophil genes induced via cytotoxic factors can result in transdifferentiation of neutrophils into CAFs [10]. A reduction in or the absence of adipocytes in diseased tissue may be a result of activated fibroblasts interfering with adipocyte differentiation [18]. In situations where adipocytes persist, they can interact with cancer cells and enable metabolic support for transdifferentiation into CAFs [17,18,20]. Carcinogenesis studies that specifically investigated pericytes in a TME did not provide strong evidence of transdifferentiation of pericytes into CAFs [18]. 

Several mechanisms have been shown to play significant roles in CAF activation. Interaction between cancer cells and fibroblasts can induce CAF phenotypes in some cancer types, such as breast cancer, through Notch signaling; however, loss of Notch signaling may promote CAF phenotypes in other cancers, such as squamous cell carcinoma [17,18]. A number of inflammatory cytokines promote CAF activation through different mechanisms; for example, IL-1 acts through NF-κB, and IL-6 acts via JAK/STAT transcription factors [21,22,33]. Interaction and positive feedback involving JAK-STATs, the contractile cytoskeleton and alterations in histone acetylation have been documented as other mechanisms that promote CAF activation [34]. Furthermore, alterations in ECM can also promote CAF activation [35,36]. Experimental trials have demonstrated that fibroblast stretching, which may develop as a result of hyperproliferation of transformed epithelial cells, may stimulate SRF-originated transcription and Yes-associated protein 1 (YAP1)–TEAD-driven transcription [18,36]. These transcription factors drive the expression of a large set of genes associated with CAFs, including the genes that express connective tissue growth factor (CTGF) and cysteine-rich angiogenic inducer 61 (CYR61) [12,36]. In addition, matricellular molecules, such as CTGF, CYR61 and the contractile cytoskeleton, crosstalk to increase tissue stiffness, which further regulates transcription programs, keeping CAFs in a self-sustaining positive-feedback loop [35]. Physiological stress can also contribute to CAF generation. Physiological and genomic stresses can promote alteration in fibroblasts. DNA breaks can induce expression of IL-6 and TGFβ [17]. The TGF-β signaling pathway has been found to have pleiotropic effects on CAF behaviors through autocrine and paracrine mechanisms [21,22]. Transgenic mice models that depleted CAFs or CAF activation signaling in tumor stromata presented direct evidence of CAFs’ protective effects against cancer [37]. Currently, due to their nonproliferative nature, senescent fibroblasts are not considered to be a major source that generates CAFs. In addition to tumor cells being a source of CAFs, signals from other cells within the HCC TME can also modulate CAF function; for example, macrophage-derived granulin promotes CAF activation in liver metastasis [38]. Furthermore, cancer therapies, including traditional chemotherapies, locoregional therapies and molecular-targeted drugs, can induce generation of CAFs and drive their functions [39]. These alterations may facilitate development of treatment resistance and occurrence of side effects [12]. Sulfatase 2 (SULF2)-induced CAFs promote HCC progression through inhibition of apoptosis and induction of an EMT in which the TGFβ1/Smad3 signaling pathway plays a critical role [40]. Furthermore, tissue inhibitor of metalloproteinase 1 (TIMP-1)-activated CAFs inhibit apoptosis through triggering of SDF1/CXCR4 signaling in HCC [41]. CAFs interact with tumor-infiltrating immune cells within the TME through effector molecules, leading to the formation of an immunosuppressive TME that provides cancer cells to evade immune surveillance from the immune system [15] Figure 2.

## 4. Cellular Origin of CAFs in HCC

With the understanding that CAFs have many subtypes, it was thought that experimental and clinical studies would provide insights into the nature and extent of CAF heterogeneity and the biological characteristics of CAFs [9]. The development of novel coculture models and implementation of single-cell RNA sequencing (scRNA-seq) techniques have provided detailed information on the origins of CAFs and revealed the presence of high levels of CAF heterogeneity in many cancer types [11,18]. Furthermore, novel genetically engineered mouse models (GEMMs) have substantially advanced our information about fibroblast origins, heterogeneity, plasticity and functions. Currently, there are no lineage-tracing models for CAFs [18]. However, intravital-microscopy imaging techniques associated with GEMMs could evaluate the origins and plasticity of activated CAF subtypes [9,17,18]. Experimental studies that utilized lecithin retinol acyltransferase-cyclization recombination enzyme (Lrat-Cre) and PDGF receptor β (PDGFRβ)-Cre have reported that hepatic stellate cells (HSCs) are the dominant source of myofibroblasts in liver cirrhosis [10,18]. Under physiological conditions, HSCs are quiescent and reside in the space of Disse. Following liver injury, HSCs lose their retinyl esters in lipid droplets, become activated and transdifferentiate into myofibroblasts that produce ECM components and αSMA [12,32]. TGFβ plays a key role in development and progression of liver cirrhosis and HCC. During hepatocarcinogenesis, TGFβ promotes transdifferentiation of HSCs into CAFs [12,42]. Another study documented that hepatocyte-derived PDGF-C could also transdifferentiate HCCs into myofibroblasts to foster progression of HCC [10,42] Figure 3.

Although the molecular basis of the cancer cell has been studied extensively, the mechanisms that activate CAFs and regulate their recruitment are only beginning to be elucidated. Many studies have demonstrated that the genetic characteristics of cancer cells play a critical role in formation of TMEs. As such, CAF signatures detected in different cancers can be utilized for stratification purposes and enable prognostic information about clinical outcomes [18,42]. Several factors have been shown to promote activation and reprogramming of CAFs, including epithelial cues such as IL-1, PDGF, metabolic reprogramming, oxidative stress, stromal cues, microRNAs and epigenetic alterations [18]. Implementation of novel genetic fate mapping and scRNA-seq techniques has enabled more favorable evidence in monitoring the origins of CAFs [10,15,16,42]. scRNA-seq provides high-resolution pictures of the transcriptomes of single cells. It is now widely used to identify cell populations within specific tissues, including that of the liver. Recently, a study that analyzed scRNA-seq data from mouse-liver cells indicated that transcription factor 21 (Tcf21) is a specific marker that distinguishes quiescent HSCs from other liver-cell types and activated HSCs [43]. Tracing the fate of Tcf21-CreER^+^ cells under normal conditions as well as after various chronic liver injuries, that study found that Tcf21-CrER^+^ preferentially marked periportal and pericentral HSCs that were quiescent in the steady state but became activated in the DEN/CCL_4_-induced state, originating 85% of the CAFs in HCC [43]. Another important finding of this study was that Tcf21-CreER^+^-targeted perivenous stellate cells are the main source of myofibroblasts and CAFs in chronically injured livers [43]. More recently, another study that utilized genetic tracing in combination with scRNA-seq analysis, as well as genetic depletion through Cr-lox-mediated deletion approaches, demonstrated that CAFs are derived primarily from resident HSCs. A second key finding of this study was that interactions between HSC-derived CAFs and tumor cells account for a main mechanism of tumor promotion and restriction in desmoplastic liver cancer, and the majority of CAFs express HSC signatures abundantly [44].

Mesenchymal stem cells (MSCs) are multipotent cells capable of differentiating into various cell types, including adipocytes, cartilage, bone and other cells, in physiological conditions [19,45,46,47,48]. MSCs have been thought to have great potential for liver regeneration and a therapeutic effect on liver fibrosis [48,49]. 

In addition to their high regenerative capacities, MSCs can be recruited within the HCC TME and exhibit significant roles in modulation of HCC progression, growth, metastasis and drug sensitivity [47,48]. They have tumor-promoting and tumor-restricting functions in HCC; however, the mechanisms that underlie these opposing effects are not fully understood [48]. In HCC patients, MSCs were demonstrated to promote apoptosis and inhibit HCC-cell proliferation, migration and invasion. Tumor-infiltrating MSCs within the HCC TME can transdifferentiate into CAF-like phenotypes after being acclimated via cancer cells [46,47]. A novel trial that investigated the impact of the HCC TME on human-adipose MSCs (hA-MSCs) and predicted hA MSC intracellular miRNA’s role demonstrated that when cocultured with Huh7 cells, the MSCs substantially upregulated the expressions of CAF markers, including αSMA, vimentin, c-MYC, MMP2, VEGF and IL-6, and thus, the hA-MSCs could transdifferentiate into CAF-like phenotypes [48]. The second key finding of that study was crosstalk between the HCC cells and components of the HCC TME to generate these CAFs [48]. Another study, which addressed the effects of the TME on differentiation of MSCs into CAFs, demonstrated that after exposure to epithelial hepatic carcinoma SK-Hep1 cells, MSCs may acquire the molecular and functional characteristics of CAFs [49]. However, the implementations of only in vitro cell-line cocultivation assays in both studies is their major limitation, and in vivo lineage-tracing experiments are required to determine transdifferentiation of MSCs into CAFs in the future [10].

Emerging evidence suggests that cancer-derived exosomes, which play a key role in carcinogenesis and tumor-cell proliferation, transdifferentiate into CAFs through a novel mechanism of endothelial-to-mesenchymal transition (EndMT) [32,50]. Epithelial cells, through EMTs or endothelial cells via EndMT, can acquire mesenchymal cell characteristics, which can be another source of CAFs [10]. Greening et al. have reported that cancer-derived exosomes induced EndMT through promotion of proliferation of endothelial cells and reconstituted premetastatic niches that formed a TME in a metastatic region [51]. They also revealed that exosome-deficient cancer cells abolish fibroblast differentiation and inhibit tumor-cell growth through silencing of exosome-secretion regulator Rab27b [51]. Cancer-derived exosomes are thought to be mediators that regulate interactions between stromal cells and reshape TMEs [51]. In another study, MSC-derived exosomes were found to inhibit EndMT, promote angiogenesis and maintain vascular homeostasis, while cancer-derived exosomes triggered EndMT, followed by induction of CAFs [50]. 

Multiple studies have reported that TGFβ1, a profibrotic growth factor, promotes adult hepatocytes to undergo phenotypic and functional properties of EMTs [10,12,17,18]. In lineage-tracing experiments that utilized AlbCre;R26Rstoplac Z double transgenic mice, researchers demonstrated that a substantial population (up to 45%) of FSP1-positive fibroblasts was derived from hepatocytes through EMTs [52]. Similar results were obtained in kidney studies that reported that approximately 40% of all fibroblasts originate via EMTs. However, this finding may be controversial; another experimental trial, which used triple-transgenic mice that expressed ROSA26 stop β-galactosidase, albumin Cre and collagen α1 green fluorescent protein (GFP), demonstrated that type-1 collagen-producing cells do not originate from hepatocytes and that hepatocytes in vivo neither acquire mesenchymal marker expression nor exhibit myofibroblast-like morphology [53]. 

In some specific conditions, HCC cells may undergo EMTs and express markers of CAFs [10,12,18]. For example, fibroblast activation protein (FAP) expression has been reported in some cancer cells as well as in CAFs, which correlates with poor clinical consequences [16,54]. FAPs can be induced under hypoxia, which is also crucial in the biological behavior of cancer cells [10,16,20,47]. Recent studies that investigated the expression levels of FAPs and hypoxia inducible factor 1α (HIF-1α) in HCC cells demonstrated that hypoxia can induce upregulation of FAPs in HCC cells and be indicative of poor prognosis in patients with HCC [10,54]. HIF-1α promotes tumor cells to acquire booster proliferation, invasion and metastasis capabilities under the metabolic stress conditions in which HIF-1α degradation is inhibited [16]. Xu et al. have reported that CAF-derived CCL5 promotes HCC metastasis through the HIF-1α/ZEB1 pathway [16]. Furthermore, they demonstrated that CCL5 was positively correlated with HIF1-α in clinical samples, and high levels of expression of HIF1-α were associated with worse overall survival [16]. CAFs secrete TGFβ, which exhibits both protumor and antitumor functions through diverse mechanisms [55]. Similar to TGFβ, nuclear liver X receptors (LXRs) either suppress or promote cancer through inhibition of cell proliferation or assistance of tumor cells in avoidance of immune surveillance [56,57]. A recent study reported that a majority of epithelial HCC cells expressed detectable LXRα levels and responded to LXR agonists, and that LXRs limit TGFβ-dependent CAF differentiation [58]. Another study, which utilized the in vitro EndMT model, documented the transdifferentiation of fetal-liver sinusoidal endothelial cells into fibroblast-like cells while mesenchymal markers were increased and the endothelial markers were decreased [59]. However, an important limitation was that the functional properties of CAFs were not investigated in these studies.

## 5. Impact of CAFs on HCC Progression

The number of studies of CAFs has increased dramatically in the last decade, with recognition that CAFs are the most significant component of the stromal cell population in a TME. CAFs are often implicated in HCC progression and drug resistance [60,61]. They modulate HCC progression through various mechanisms, including direct effects on HCC cells through secretion of soluble factors and exosomes and indirect effects through other stromal cells and ECM remodeling [21,23]. Recently, based on scRNA-seq data from the TCGA and GEO databases, Yu et al. identified four CAF subpopulations in HCC, three of which have been associated with prognosis in patients with HCC [62]. Of the total four hundred and twenty-three analyzed genes, six were primarily linked with 39 pathways, such as those of angiogenesis, apoptosis and hypoxia. Another significant finding of this trial was that risk signatures were substantially associated with stromal and immune scores, as well as some immune cells. In another study, Qi et al. showed that CAF-derived exosomal miR-20a-5p facilitated HCC progression through the LIM domain and actin binding 1 (LIMA1)-mediated β-catenin pathway [63] Table 1.

Previous studies have revealed that CAFs promote cancer progression and metastasis through production of a variety of soluble factors, including inflammatory cytokines, growth factors and chemokines [12,16,19,20,21,22,23,64]. However, CAFs are phenotypically and functionally heterogeneous and can exhibit both protumorigenic and antitumorigenic activity [65,66,67,68]. A recent trial that implemented proteomics and sc-RNS-seq analysis in order to examine the CAF landscape in HCC demonstrated three major CAF populations in HCC, one of which specifically expresses the prolargin protein that binds and inhibits activity of several proangiogenic proteins, including hepatocyte and fibroblast growth factors. As such, prolargin is thought to be an angiogenesis modulator and a CAF-derived tumor suppressor in HCC [68]. Studies that primarily investigated the Hedgehog (Hh) signaling pathway indicated that CAFs could also have antitumoral activity in some conditions [12,16,17,18]. CAFs with specific proteomic profiles can exhibit a tumor-inhibitory function. A trial that addressed stromal transcriptional signatures in some tumors showed the presences of various stromal transcriptional signatures in tumor tissue samples, and aggressive tumors were associated with distinct stromal signatures [15,16,17,18]. More recently, Song et al. reported detailed cytokine-regulated crosstalk between CAFs, HCC cells and TANs, fostering tumor-cell migration and invasion [69] Figure 4. 

In 2011, Mazzocca et al. demonstrated, for the first time, the existence of crosstalk between CAFs and HCC cells [70]. They indicated the molecular and functional differences between peritumoral fibroblasts (PTFs) and CAFs [70,71]. Additionally, they showed that HCC-derived lysophostatidic acid (LPA) plays a critical role in promotion of transdifferentiation of PTFs to CAF-like myofibroblastic phenotypes, which in turn accelerates HCC progression [70]. Similar to stem cells, tumor-initiating cells are orchestrated through various signals generated within their TMEs. Lau et al. revealed that CAF-derived hepatocyte growth factor (HGF) orchestrates tumor-initiating cell plasticity in HCC through activation of c-Met/FRA1/HEY1 signaling [30]. Furthermore, they found that HGF-induced FRA1 activation was associated with the fibrosis-dependent development of HCC in a STAM NASH-HCC mouse model [30]. Another significant finding of this study was that the presence of αSMA-positive CAFs correlated with worse clinical consequences [30]. Another trial, which used an in vitro model of paracrine interaction between HCC-cell lines (HepG2, SNU423) and HSC and investigated the regulatory mechanism that underlies keratin 19 (KRT19) expression in HCC, demonstrated that KRT 19 expression in HCC is orchestrated through fibroblast-derived hepatocyte growth factor (HGF) via a MET-ERK1/2-AP1 and SP1 signaling pathway [72]. 

Previous studies have shown that CAFs are closely related to invasion and metastasis of HCC cells, but the mechanisms of CAFs that drive HCC metastasis were not completely clarified [73,74]. CAFs do not exist independently around tumors, but crosstalk with tumor cells to promote their malignant phenotypes [73,74]. Tumor cells can recruit CAF precursors and transdifferentiate normal fibroblasts into CAFs. Meanwhile, CAFs secrete large amounts of cytokines, chemokines, growth factors and ECM proteins, which form TMEs to promote HCC-cell proliferation, metastasis and drug resistance [16,75,76]. The chemokine–chemokine receptor (CK-CKR) network represents a key regulator of immune-cell recruitment and shapes the TME [77]. Studies have indicated that CAFs upregulate levels of CCL2, CCL5, CCL7, CCL26 and CXCL17 and acquire a booster-tumor metastatic phenotype [23]. CCL7 and CXCL16 promote both migration and invasion of HCC cells, while CCL2 and CCL5 promote only migration of HCC cells [78]. Moreover, CCL2 and CCL5 activate the Hh signaling pathway, while CCL7 and CXCL16 boost the activity of TGFβ in HCC cells. More recently, Xu et al. found that CCL5 was the most significant cytokine in the CAFs that promote HCC metastasis; they also observed that serum CCL5 levels were quite high in patients who developed HCC in cirrhotic livers [16]. CCL5, an inflammatory cytokine, plays a relevant role in CAF promotion of carcinogenesis [79]. Hypoxia-inducible factor 1 alpha (HIF1α) exhibits crucial roles in regulation of energy metabolism, angiogenesis and other processes in the TME and provides cancer cells to gain more proliferation and metastasis capacity. It has been reported that CAFs have been involved in regulation of tumor HIF1α to promote tumor progression [16,17,18,42]. Xu et al. indicated that CAF-derived CCL5 inhibited HIF1α ubiquitination degradation, maintained HIF1α expression under normoxia and promoted an EMT and metastasis through activation of downstream factor ZEB1 [16]. Meanwhile, CCL5 was positively correlated with HIF1α in clinical samples; these high expressions were significantly associated with poor prognosis [16].

Recently, CAFs have been documented to recruit immune cells within the TME, including neutrophils and macrophages [80,81]. Endosialin is a transmembrane protein that is expressed in some cancer cells, stromal cells and pericytes, but barely expressed in normal tissue. In a novel study, Yang et al. documented that endosialin is particularly expressed in CAFs in HCC and its expression is associated with worse overall survival in HCC patients [82]. They also observed that endosialin could regulate expression of growth arrest-specific protein 6 (GAS6] in CAFs, which promotes M2 polarization of macrophages to promote HCC progression [82]. Additionally, another trial that investigated the impact of HCC-derived CAFs on neutrophils revealed that the HCC-derived CAFs induced chemotaxis in the neutrophils and protected them from spontaneous apoptosis [83]. These researchers found that the HCC-derived CAFs promoted activation of STAT3 pathways in the neutrophils, which was essential for the survival and function of the activated neutrophils [83]. One of the significant findings of this study was that HCC-derived CAFs primed neutrophils’ impaired T-cell function through the PD1/PDL1 signaling pathway [83]. It has been suggested that HCC-derived CAFs regulate survival, activation and function of neutrophils within HCC through an IL6-STAT3-PDL1 signaling pathway, which represents a novel mechanism for the role of CAFs in remodeling the cancer niche and provides a potential target for HCC therapy [15]. 

**Table 1 ijms-24-03941-t001:** Impact of CAFs on HCC progression.

Authors	Type of Trial	Signaling Pathways, Mediators and Key Findings	Reference
**Mazzocca et al., 2011**	Clinical/Experimental	HCC cells secrete lysophostatidic acid (LPA), which promotes transdifferentiation of peritumoral tissue fibroblasts (PTFs) into CAFs HCC-secreted LPA accelerates HCC progression through recruitment of PTFs and promotion of their differentiation into myofibroblasts Higher serum levels of LPA are associated with worse survival	[70]
**Lau et al., 2016**	Clinical/Experimental	The presence of α-SMA^+^ CAFs correlates with poor clinical outcomes CAF-derived HGF regulates liver tumor-initiating cells (T-ICs) through activation of FRA1 in an ERK1/2-dependent manner HGF-induced FRA1 activation was associated with fibrosis-dependent development of HCC in a STAM NASH-HCC mouse model	[30]
**Rhee et al., 2018**	Experimental	Keratin 19 (KRT19) expression in HCC is regulated via crosstalk between CAFs and HCC cells through a MET-ERK1/2-AP1 and SP1 pathway HSCs upregulate transcription and translation of KRT19 in HCC cells via paracrine interactions HSC-derived HGF activates c-MET and the MEK-ERK1/2 pathway, which upregulates KRT19 expression in HCC cells In HCC specimens, HGF and KRT19 protein expression correlated with CAF levels	[72]
**Zhang et al., 2017**	Experimental	A significant reduction in the miR-320a level in CAF-derived exosomes Stromal cells could transfer miR-329a to HCC cells The miR-320a-PBX3 pathway inhibits HCC progression through suppression of activation of the MAPK pathway CAF-mediated HCC progression is partially related to loss of antitumor miR-320a in the exosomes of CAFs	[74]
**Affo et al., 2017**	Clinical/Experimental	CAFs play a key role in development and progression of HCC	[76]
**Xu et al., 2022**	Clinical/Experimental	CAF-derived chemokine CCL5 enhances HCC metastasis through triggering of the HIF1α/ZEB1 pathway HCC-derived CAFs promote migration and invasion of HCC cells and boosted metastasis to the lungs of NOD/SCID mice CAF-derived CCL5 inhibits ubiquitination and degradation of hypoxia-inducible factor 1 alpha (HIF1α) under normoxia, thereby upregulating the downstream gene zinc finger enhancer-binding protein 1 (ZEB1) and inducing epithelial mesenchymal transition (EMT)	[16]
**Yang et al., 2020**	Experimental	Endosialin is mainly expressed in CAFs in HCC, and its expression inversely correlates with patient prognosis Endosialin interacts with CD68 to recruit macrophages and regulates expression of GAS6 in CAFs to mediate M2 polarization of macrophages Endosialin-positive CAFs promote HCC progression	[82]
**Cheng et al., 2018**	Experimental	HCC-CAFs induce chemotaxis of PDL1^+^ neutrophils through the IL6-STAT3 pathway that boosts immune suppression in HCC	[83]
**Song et al., 2021**	Clinical/Experimental	CAF-derived cardiotrophin-like cytokine factor 1 (CLCF1) increases chemokine (C-X-C motif) ligand 6 (CXCL6) and TGF-β secretion in HCC cells, which promotes HCC-cell stemness and TAN infiltration and polarization HCC-derived CXCL6 and TGF-β activate ERK1/2 signaling of CAFs to produce more CXCF1 and promote HCC progression Selective blocking of CLCF1 or ERK1/2 signaling could provide an effective therapeutic target for HCC patients	[69]
**Qi et al., 2022**	Experimental	CAF-derived exosomal miR-20-a-5p promotes HCC progression through the LIMA1-mediated Wnt/β-catenin signaling pathway LIMA1 is downregulated via CAF-derived exosomes that carry oncogenic miR-20a-5p in HCC	[63]

Although interaction between HCC cells, CAFs and other stromal cells in the HCC TME has been well documented, the nature of the complex interaction between CAFs and other components within the TME has not yet been completely elucidated, mainly during the distinct HCC stage [71]. Recently, Song et al. demonstrated complex interactions between HCC cells, CAFs and TANs, which enhance cancer stemness and recruitment of TANs in HCC [69]. They reported, for the first time, that CAFs isolated from advanced-stage HCC showed a greater tumor-promoting effect in vivo than those isolated from the early stages [20]. This study highlighted the clinical significance of CAF-derived cardiotrophin-like cytokine factor 1 (CLCF1) signaling in CAF-mediated direct crosstalk with tumor cells and indirect interaction with TANs within the HCC TME [69]. Furthermore, the researchers thereof reported that CAF-derived CLCF1 upregulated two key cytokines, chemokine (C-X-C motif) ligand 6 (CXCL6) and TGFβ, in HCC cells through the Akt/extracellular signal-regulated kinase 1/2 (ERK1/2)/STAT3 signaling pathway, which in turn induced HCC-cell stemness and TAN infiltration and polarization [69]. Clinically, high levels of CLCF1 expression have been found to be correlated with aggressive tumor behaviors and worse clinical outcomes [69]. In HCC, tumor cells secrete a large amount of TGFβ, while CAFs produce relatively lower levels of TGFβ, IL-6 and granulocyte-colony-stimulating factor (GCSF) to promote N2 neutrophils [15,69,71]. These data clearly indicate that CAFs predominantly induce HCC progression via paracrine pathways [69,71]. However, other mechanisms through which CAFs orchestrate HCC progression, including exosomes and extracellular vesicles, can play a role in this progression [15,17,18,69,71]. CLCF1-promoted CXCL6 and TGFβ constitute the crucial bridge that connects cellular crosstalk between CAFs, HCC cells and TANs [69]. The study thereof reported that CXCL6 fosters HCC stemness via transcriptional driving of E2F1 [69,71]. Additionally, microRNA (miRNA) that is involved in E2F1 dysregulation exhibits a critical role in tumor progression, and CXCL6 influences miRNA function in carcinogenesis [69,71].

Neutrophils are innate immune cells that are thought to be a double-edged sword during carcinogenesis [15,83,84,85,86,87,88]. Distinct TMEs polarize neutrophils to antitumorigenic (N1) or protumorigenic (N2) phenotypes [15,83,84]. N2 TANs have the capacity to form neutrophil extracellular traps (NETs), which can act to promote HCC development in the setting of cirrhosis [15,84]. Emerging evidence reveals that high infiltration levels of tumor-associated neutrophils (TANs) within the TME are correlated with worse overall survival in some solid tumors [15]. TANs promote cancer progression through induction of tumor-cell proliferation, metastasis and stemness; remodeling of the ECM; augmentation of angiogenesis; or stimulation of immunosuppression. In HCC [15,69,85,88,89,90,91], TANs have been reported to increase tumor-cell stemness and recruit immunosuppressive macrophages and Tregs [88]. However, their exact role in hepatocarcinogenesis and the effects of TMEs on education about TANs, regarding their phenotypes and functions, are largely unknown. Multiple studies have highlighted the relevance of interaction between HCC cells, TANs and CAFs in affecting HCC progression [15,17,18,69]. CAFs can suppress neutrophil function through the SDF1a/CXCR4/IL-6 pathway, which promotes production of CD66b, PD-L1, CXCL8/IL-8, TNF and CCL2, which can inhibit the function and proliferation of T cells in vitro [82]. CAF-derived CLCF1 can promote tumoral expression of CXCL6 and TGFβ, resulting in neutrophil recruitment and N2 polarization, respectively [69]. TANs acquired a protumoral N2 phenotype in the middle and advanced stages in correlation with increased levels of CLCF1 [69]. In advanced HCC, high levels of CLCF1 expression result in an augmented CLCF1-CXCL6/TGFβ pathway, recruiting more TANs and polarizing them toward the N2 phenotype to further facilitate tumor progression [15,69,71]. As such, CLCF1 may be a potential prognostic biomarker for HCC, and selective blocking of CLCF1 signaling could provide an effective therapy for HCC patients [15].

## 6. Targeting CAFs for Clinical Benefits

After documentation of the correlation between CAF numbers and CAF functions and clinical outcomes, a number of CAF-targeting strategies were investigated in preclinical and clinical studies [10,15,17,18,81]. The rationale for targeting CAFs is to reduce ECM and immunosuppressive ligands in order to increase the efficacy of anticancer approaches [15,18,81]. Other targeting mechanisms, such as a TGFβ signaling pathway that activates CAFs to regulate tumor phenotypes, are being extensively investigated [17]. However, the breadth of CAF functions and the ability to transdifferentiate subtypes into each other pose a challenge in this field [10,15,17,18,21,81]. Furthermore, preclinical studies have shown that nonspecific targeting or deletion of stromal fibroblasts is ineffective in tumor control [17]. As such, targeting CAF subtypes or reprogramming CAFs into normal fibroblasts or antitumorigenic phenotypes may improve clinical outcomes [17]. Making CAFs more normal could be an attractive strategy; for example, treatment with vitamin D receptor ligands in pancreatic cancer reverted activated stellate cells into a quiescent state and controlled the disease’s progression [17,18]. In this regard, it is important to determine whether individual fibroblast populations account for “states” and are interconvertible or whether different “lineage-restricted” effects occur [17]. Contributions of CAF functions to tumor biology are assumed to differ between tumor types, but this has yet to be elucidated.

In clinical practice, neither elimination nor reprogramming of CAFs is required to achieve clinical benefit, but they can be achieved through blocking of CAF-derived signals [17]. In many tumors, CAFs are the major sources of chemokines, which induce chemotherapy resistance through expression of ECM components that mitigate access to drug delivery through creation of a booster barrier and compression of blood vessels and lymphatic vessels, which result in hypoperfusion [17,18,91,92,93]. Targeting ECM components and downstream signaling may be an approach to interfering with CAF–cancer cell communication [92]. Among approaches that deplete the ECM, PEGPH20 targets the ECM component hyaluronan (HA) [92,93]. In clinical trials, PEGPH20 and chemotherapy initially showed positive results, but recently failed in a phase-III study [18,93]. SMO inhibitors, which promote ECM depletion through suppression of the Hh signaling pathway and targeting of ECM-producing αSMA^+^ CAFs, have not demonstrated effects for PDAC [18,94]. Rho-associated protein kinase inhibitors, such as Fasudil and AT13148, and antibodies against ECM proteins, such as connective tissue growth factor (CTGF), fibronectin and tenascin C, are among other strategies to target ECMs. Blocking CTGF in pancreatic cancer modulates cancer-cell survival cues, enhancing chemotherapy responses [94,95]. These data indicate that targeting ECM proteins can be therapeutically effective [17,18,93,94]. It has been observed that CAFs lose their fibroblastic properties and become deactivated during chronic hypoxia, indicating that CAFs may transdifferentiate into a quiescent phenotype [95,96]. Treatment approaches that transdifferentiate activated CAFs into this quiescent phenotype include all-trans retinoic acid (ATRA), minnelide and the vitamin D receptor agonist calcipotriol, which facilitate resolution of liver and pancreatic fibrosis and enhance pancreatic cancer therapy [18,97]. Furthermore, losartan, the angiotensin receptor II antagonist, was reported to reduce TGFβ activation in αSMA^+^ CAFs, resulting in a decrease in desmoplasia and an increase in drug delivery [98]. 

Recent advances regarding the molecular and functional heterogeneity of CAFs are seminal in cancer treatments. Potential strategies for development of novel therapies would be either to target CAF-derived tumor promotion and immunosuppressive ligands, such as IL-6 and TGFβ, or to inhibit subtype-specific signaling pathways that would destroy specific CAF populations [18,99,100]. Another strategy would be to design treatments that take advantage of the plasticity of CAFs and shift tumor-promoting CAF subtypes to the quiescent phenotype [18]. Finally, only combinatorial strategies that target CAFs, cancer cells, immune cells and drug delivery may be successful as effective treatments [18]. Emerging evidence suggests that cancer cells and CAFs share certain signaling pathways, which can be a guide to development of therapies that target cancer cells without affecting stromata [17,18]. Furthermore, treatments that specifically target the protumorigenic function of CAFs can be designed. For instance, in addition to pancreatic cancer cells, the IL-1 receptor antagonist anakinra may also target CAFs with potential tumor-promoting and immunosuppressive effects [101,102,103,104,105]. However, the existence of overlapping signaling pathways among distinct cell types is an issue to be thought of in design of combinatorial therapies [105,106]. For example, JAK inhibitors are effective at targeting cancer cells and CAFs while also targeting proliferation and activity of CD8^+^ T cells [18,105,106,107]. As such, imaging, genetic and immunohistochemical examinations will be required to be included in planning of clinical studies to identify influences of anticancer drugs, including molecular-targeted drugs, on CAFs and the TME. Only combinatorial therapeutic approaches that consider these issues and target the tumor-promoting components of the TME are likely to achieve this (Table 2).

## 7. Present Challenges and Future Direction

In recent years, emerging studies have highlighted that cancers greatly depend on their surrounding TMEs and that CAFs within those TMEs are critical for cancer development and progression because of their diverse roles in ECM remodeling; maintenance of stemness; blood-vessel formation; modulation of tumor metabolism; immune responses; promotion of cancer-cell proliferation, migration and invasion; and therapeutic resistance. Utilization of scRNA-seq and gene-lineage tracing techniques has identified HSCs to be the main source of CAFs and also indicated multiple levels of molecular and functional CAF heterogeneity. However, scRNA-seq analysis that identifies subpopulations of CAFs cannot completely determine their protein expression profiles. One of the challenges in defining CAF heterogeneity with scRNA-seq analysis is that population subclustering can be arbitrarily defined and is limited by the number of samples. Another issue is that CAFs are difficult to isolate. Therefore, CAFs are generally less represented in scRNA-seq data sets, and tissue-specific protocols should be developed. As such, other techniques are required to explore distinct functions of CAF subpopulations. Research of CAFs has been made possible through their ability to be cultured in vitro, but the cell-culture process and the exact conditions thereof can affect cell behaviors. In the future, relevant information will come from studies that focus on optimizing in vitro coculture models and creating in vivo lineage-tracing models to investigate the functions and origins of CAFs.

The biological activities of CAFs are mediated through various intracellular and extracellular factors, especially those in signaling pathways, such as the TGFβ, JAK/STAT, NF-κB, MAPK and Wnt/β-catenin pathways, which are closely related to tumor progression. These signaling pathways exhibit their own special characteristics during cancer progression and have the potential to be targeted for anticancer therapy. Since CAFs exert molecular and functional heterogeneity in different cancers and because of the specific crosstalk between CAFs and cancer cells, specificity and diversity of CAFs should be considered for optimal therapeutic efficacy in development of treatment strategies. CAFs exert both protumoral and antitumoral activities in HCC; as such, their precise functions should be determined before CAF-targeted therapies are started. Several clinical studies of CAFs have shown that they have a promising future in cancer treatment; however, there are also multiple hurdles that need to be overcome before CAFs can be targeted in cancer treatment. Targeting TMEs has been an appealing implementation and had a bumpy history, with failures in the area of MMP inhibition, mixed results from angiogenesis targeting and transformative results with immune checkpoint inhibitors in some cancers. Therefore, a comprehensive understanding of signaling pathways that mediate crosstalk between CAFs and cancer cells is required to fully realize their pivotal roles and to translate favorable findings from CAF studies into clinical benefits.

## Figures and Tables

**Figure 1 ijms-24-03941-f001:**
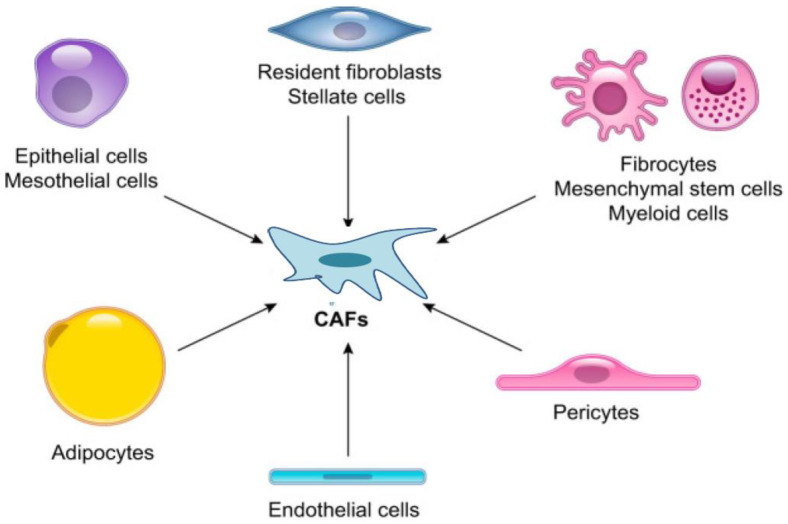
Epithelial cells, mesothelial cells, resident fibroblasts, pancreatic and hepatic stellate cells, pericytes, adipocytes, mesenchymal stem cells, myeloid cells and endothelial cells have been reported as potential cellular origins of cancer-associated fibroblasts (CAFs).

**Figure 2 ijms-24-03941-f002:**
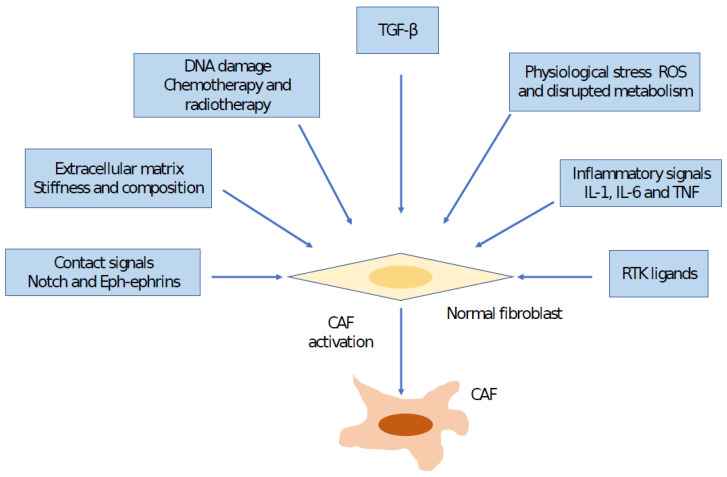
This figure indicates the mechanisms involved in cancer-associated fibroblast (CAF) activation. FGF, fibroblast growth factor; PDGF, platelet-derived growth factor; ROS, reactive oxygen species; RTK, receptor tyrosine kinase; TGFβ, transforming growth factor β; TNF, tumor necrosis factor.

**Figure 3 ijms-24-03941-f003:**
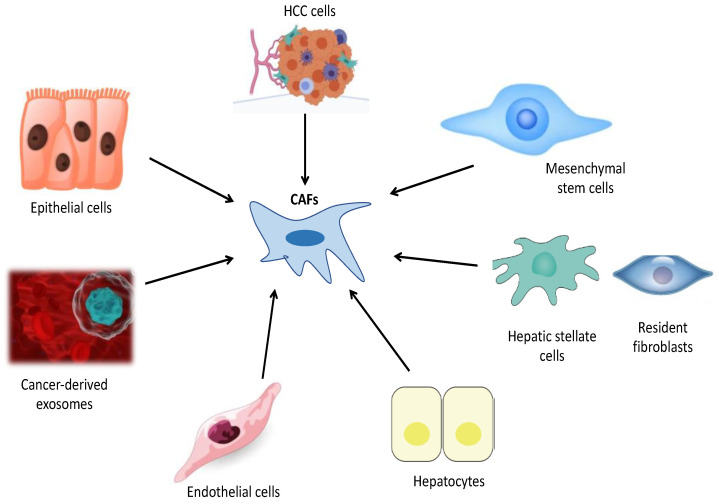
This figure indicates potential cells from which cancer-associated fibroblasts (CAFs) originate in HCC: HCC cells, Mesenchymal stem cells, hepatic stellate cells, resident fibroblasts, hepatocytes, endothelial cells, cancer-derived exosomes and epithelial cells.

**Figure 4 ijms-24-03941-f004:**
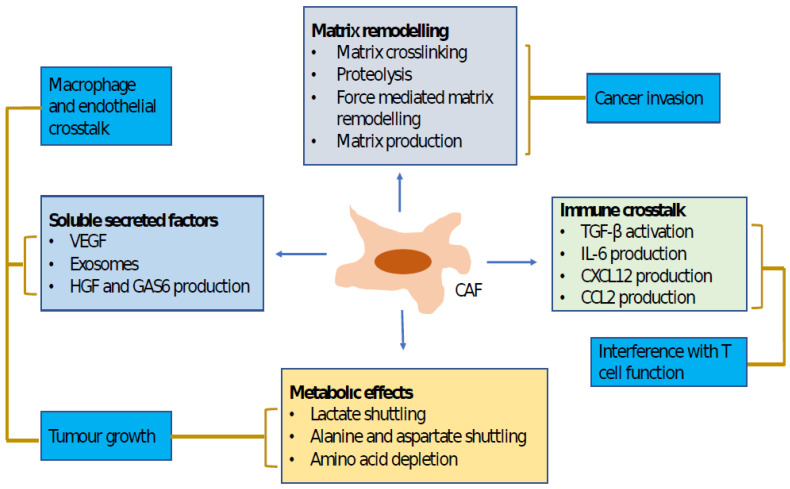
This figure indicates cancer-associated fibroblast functions and the mechanisms that regulate them. Lines connect mechanisms to functions. Both matrix remodeling and production of soluble factors promote tumor-cell invasion. Soluble factors also play a critical role in tumor-cell growth and changes in tumor microenvironments, which are also influenced by the altered metabolic states of tumors. CAF, cancer-associated fibroblast; CCL2, CC-chemokine ligand 2; CXCL12, CXC-chemokine ligand 12; IL-6, interleukin 6; GAS6, growth arrest-specific protein 6; HGF, hepatocyte growth factor; TGFβ, transforming growth factor-β; VEGF, vascular endothelial growth factor.

**Table 2 ijms-24-03941-t002:** Clinical trials targeting cancer-associated fibroblasts.

Target	Name	Drug and Biologic	Mechanism	Current Status	References
Inhibition of CAF activation					
FGFR	JNJ-42756493	Small-molecule inhibitor	Promotes ECM depletion Prevents CAF activation	Phase-I and phase-II trials ongoing	[93]
Hedgehog signaling pathway	JPI-926 (saridegib) and vismodegib	Small-molecule inhibitor	Leads to ECM depletion via blocking the Hedgehog signaling pathway and targeting ECM-producing αSMA^+^ CAFs	Failed in phase-III trial	[92]
Inhibition of CAF activation and CAF action					
TGF-β	Various, including galunisertib	Both blocking Abs and small-molecule receptor inhibitors	Prevents CAF activation and immunosuppression	Phase-II and phase-III trials ongoing	[94,95]
Anjiotensin receptor II antagonist	Losartan	Small-molecule inhibitor	Decreases TGF-β activation in αSMA^+^ CAFs, leading to an increase in drug delivery and immunotherapy efficacy	Phase-III trial ongoing	[99]
Inhibition of CAF action					
CXCR4	AMD3100	Small-molecule inhibitor	Prevents signaling from CAFs to immune cells	CTs ongoing	[96]
ROCK	AT13148	Small-molecule inhibitor	Reduces contractility	Phase-I trial completed	[100]
FAK	Defactinib	Small-molecule inhibitor	Reduces downstream signaling of integrins	Clinical trials ongoing	[92]
LOXL2	Simtuzumab	Blocking Ab	Anticrosslinking	Preclinical and fibrosis trials ongoing	[102]
CTGF	FG-3019	Blocking Ab	Blocks binding to receptors, including integrins	Early-phase clinical trials ongoing	[101]
Hyaluronic acid	PEGPH20	Pegylated enzyme	Degrades ECM to increase access and efficacy of anticancer treatments and immunotherapies	Failed in phase-III trial	[97,98]
FAP-express cells	Various, including PT630 and RO6874281	Blocking Abs, molecular radiotherapy, inhibitors or an Ab-IL-2 fusion	Blocks FAB CAF function, promoting T-cell function	Phase-I and phase-II trials under way	[105]
CAF normalization					
Vitamin A metabolism	ATRA minnelide	Vitamin A metabolite	Transdifferentiates aHSCs into qHSCs	Clinical trials ongoing	[106]
Vitamin D receptor	Calcipotriol	Small-molecule agonist	Transdifferentiates aHSCs into qHSCs	Clinical trials ongoing	[107]

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
