# Peer review of "Emerging Role of Cancer-Associated Fibroblasts in Progression and Treatment of Hepatocellular Carcinoma"

_ijms, 2023, doi:10.3390/ijms24043941_

Round 1

Reviewer 1 Report

The current manuscript is a review article claiming a comprehensive discussion on molecular mechanisms involved in crosstalk between CAFs, HCC cells and other stromal cells as well as the effects of CAFs on HCC cell growth, metastasis, drug resistance and clinical outcomes.

Overall, the review has summarized the research on CAF in HCC in last decades.

However the presentation needs revised writing focusing on the following points:

·         Line 1 and 39 has disparity in occurrence and mortality condition.

·         In lines 94-97 some specific secreted factors are mentioned. Specific molecules involved (ref 16, 19 ) are unnecessary in introduction section.

·         Elaborate introduction to Fibroblasts (Line 117 onwards) is unnecessary and repetitive in this context. Elaborating on CAF cell types of origin as well as general and HCC specific features of CAFs should suffice with a brief note on Fibroblast morphology, markers, and function.

·         The section “Origin and Activation Mechanisms of CAFs ” is well written. However, Additional summarized table focusing on reported and/or predicted correlation on cell type of origin and activation mechanisms would enhance lucidity of the manuscript.

·         Heterogeneity, plasticity, and function of CAFs reported in multiple publications are put together in “Cellular Origin of CAFs in HCC” section. However, important citations at the right place are missing and it is hard to interpret the information.

For improving the readability of this section, a summary chart or diagram would be helpful where the cell types are well categorized according to their diversity and functionality.

·         MSC multipotency can be mentioned as in lines 280-282, but only after correctly cited mention of MSC derived CAFs in HCC context.

·         The section of “Impact of CAFs on HCC Progression” has mentioned a large number of scietific reports but a summary table would better resolve the overall information where the author can identify and highlight on the types of signaling mechanisms and corresponding subfamily of mediators reported so far.

·         For similar concerns, a table in the “Targeting CAFs for Clinical Benefits” section, summarizing drug types and targeting mechanisms would improve the readability of the paper.

·         doi: 10.3390/ijms20071723. Important updated review. Please consider this review and cite if possible.

·         doi: 10.3390/cells11233857 ;  doi: 10.3892/ijo.2021.5239 ; doi: 10.1111/cas.14262;  doi: 10.3389/fimmu.2022.1009789 ; doi: 10.1111/cas.15041 ; doi: 10.1038/s41388-021-02171-z  ……………………………and more: some important recent updates are missing.

Addressing these concerns could be beneficial for the acceptance of the manuscript for publication.

Author Response

Author’s Reply to the Review Report (Reviewer 1)

First of all, I would like to thank you for your valuable comments regarding the manuscript. I made corrections in line with your recommendations and completed the deficiencies. You can find them below,

  • I corrected the disparity related to mortality due to HCC in lines 1 and 39

  • I removed some specific molecules, such as CCL2, CCL5, CCL7, and CXCL16 secreted by CAFs in lines 94-97

  • In the paragraph starting with line 188 (according to the previous manuscript in line 117) in ‘’ Origin and Activation Mechanisms of CAFs’’ section, I mentioned mainly neutrophils as the origin of CAFs and the mechanisms involved in the transdifferentiation of neutrophils to CAFs, as well as other cells from which CAF derives.

  • The previous manuscript contained three figures, two of which pertained to the potential origin and activation mechanisms of CAFs. In addition to these figures, should I prepare a table on the subject? Could you please inform me about this?

  • I must say that I have benefited greatly from the references you suggested. I have used the knowledge contained in the references in the relevant sections of the manuscript and placed the references. I placed correct references you suggested in the ‘’ Cellular origin of CAFs in HCC’’ section, Also, I prepared a figure demonstrating cells from which CAFs originate in HCC context (Figure 4).

  • I have supplemented the knowledge on MSC multipotency in lines 298-301, in ‘’ Cellular origin of CAFs in HCC’’ section.

  • I have prepared a table containing signaling pathways, mediators and key findings regarding the ‘’ Impact of CAFs on HCC progression’’ section (Table 1).

  • I have prepared a table of drug types and mechanisms regarding the ‘’ Targeting CAFs for clinical benefit’’ section (Table 2).

Kind regards,

Hikmet Akkız

Reviewer 2 Report

I have gone through the topic. It is indeed interesting and the concepts are clearly illustrated.  cell signaling pathways have contributory roles. However, I will encourage the author to first describe the role of cell pathways in the introduction section. Similarly, brief overview of TGF/SMAD, JAK/STAT will add more attractive information to the review. Likewise role of CAFs in mice needs more explanations. How CAFs behaved in tumor-bearing models and is there any evidence of metastatic spread of cancer cells in mice inoculated with CAFs? Concluding remarks should be discussed more comprehensively in context of cell signaling pathways role in the interplay between CAFs and tumor microenvironment. 

Author Response

Author’s response to the reviewer 2

I would like to thank you for your valuable comments about my manuscript. I made additions based on your suggestions.

You can find them below,

  • In the 3rd paragraph of the ‘’Introduction’’ section of the manuscript, I mentioned about the signaling pathways regulating crosstalk between CAFs and HCCs, including TGFb/SMAD and JAK/STAT signaling pathways.

  • I have provided information on the effect of CAFs on tumor progression in mice on line 230-232 (reference 38), in ‘’ Origin and Activation Mechanisms of CAFS’’ section.

  • Unfortunately, I could not find an article on metastasis of cancer cells in CAF-inoculated mice.

  • In the ‘’Present Challenges and Future Direction’’ section , I highlighted in more detail the role cellular signaling pathways in the crosstalk between CAFs and cancer cells.

Kind regards,

Hikmet Akkız

Reviewer 3 Report

This review summarized the present publications and frontiers of research in CAFs, and provided the background and basic mechanisms of CAFs, related with HCC progression and drug resistance. Meantime, authors showed the potential target therapy to CAFs for HCC and provided a future research focus. 

There I recommended accepting this MS.

Author Response

Author’s response to the reviewer 3

I would like to thank you for your valuable comments about my manuscript. I made additions you suggested. You can find them below,

  • In section of Cellular Origin of CAFs in HCC of the manuscript, I mentioned the SULF2 (reference 41) and TIMP-1 (reference 42) mechanisms that play a role in CAF activatio

  • I also gave information about the crosstalk between CAFs and immune cells within the TME in the same section (reference 15).

  • I have prepared a table of CAFs targeted therapies (Table 2)

  • The limitations of CAF studies are mentioned in several sections of the manuscript, for example in ‘’Cellular origin of CAFs in HCC’’ section

Kind regards,

Hikmet Akkız

Reviewer 4 Report

1.      Please mention other factors and pathways for CAF activation. For example, SULF2 (doi:10.3389/fcell.2021.631931), CTHRC1 (doi: 10.1016/j.ebiom.2019.01.009), TIMP-1 (doi: 10.18632/oncotarget.3616).

2.      Please allude to the interactions between CAFs and tumor-infiltrating lymphocytes in the tumor microenvironment of HCC. For example, about the immune cells involved, CAFs-secreted factors, mechanisms, etc.

3.      Please represent a summary of CAF-targeted treatments in a table.

4.      Please state about the limitations of studying CAFs.

Author Response

Author’s response to the reviewer 4

I would like to thank you for your valuable comments about my manuscript. I made additions you suggested. You can find them below,

  • In section of Cellular Origin of CAFs in HCC of the manuscript, I mentioned the SULF2 (reference 41) and TIMP-1 (reference 42) mechanisms that play a role in CAF activatio

  • I also gave information about the crosstalk between CAFs and immune cells within the TME in the same section (reference 15).

  • I have prepared a table of CAFs targeted therapies (Table 2)

  • The limitations of CAF studies are mentioned in several sections of the manuscript, for example in ‘’Cellular origin of CAFs in HCC’’ section

Kind regards,

Hikmet Akkız

Round 2

Reviewer 1 Report

The authors have responded to the comments made in last revision.

However I could not find Table 1 and Table 2 in the the manuscript or supplemental documents .

Figure 3 title should specify CAF origin from HCC. 

Author Response

Dear Reviewer,                                                                                   January 7, 2023

I am so sorry for the inconvenience.

I had sent the revised manuscript including tables to Dr. Qiu on December 29, 2022. Dr. Qiu stated that I could send the revised manuscript to him and he would upload it the system.

I have changed the title of Figure 3.

I have uploaded the revised manuscript.

Kind regards,

Hikmet Akkız

Reviewer 4 Report

Thank you for the revisions.

Author Response

Thank you 

Round 3

Reviewer 1 Report

The manuscript can be accepted in its current form.